# Exploring the Impact of Supervised Multimodal Learning on the Performance and Explainability of Pediatric Brain Tumor Molecular Diagnosis

Sara Ketabi[1,2,3], Matthias W. Wagner[6,7], Cynthia Hawkins[9], Uri Tabori[8], Birgit Betina Ertl-Wagner[1,4,6,10], and Farzad Khalvati[1,2,3,4,5,6,10*]

[1]Neurosciences and Mental Health Research Program, The Hospital for Sick Children, Toronto, Canada
[2]Department of Mechanical and Industrial Engineering, University of Toronto, Toronto, Canada
[3]Vector Institute for Artificial Intelligence, Toronto, Canada
[4]Institute of Medical Science, University of Toronto, Toronto, Canada
[5]Department of Computer Science, University of Toronto, Toronto, Canada
[6]Department of Diagnostic and Interventional Radiology, The Hospital for Sick Children, Toronto, Canada
[7]Department of Diagnostic and Interventional Neuroradiology, University Hospital Augsburg, Augsburg, Germany
[8]Division of Hematology and Oncology, The Hospital for Sick Children, Toronto, Canada
[9]Paediatric Laboratory Medicine, Division of Pathology, The Hospital for Sick Children, Toronto, Canada
[10]Department of Medical Imaging, University of Toronto, Toronto, Canada
*farzad.khalvati@utoronto.ca

*Abstract*—Pediatric Low-grade Glioma (pLGG) is one of the most common brain tumors in children, and identifying its genetic markers (molecular diagnosis) is crucial for tumor prognosis and targeted treatment planning. Convolution Neural Networks (CNNs) have shown strong performance in predicting these genetic markers from brain magnetic resonance imaging (MRI) data. Nonetheless, most CNN-based architectures rely on tumor segmentation masks to specify regions of interest (ROIs) in the image to achieve high performance, which are labor-intensive and costly to obtain. In this work, we propose a supervised multimodal learning framework that integrates radiology reports, readily available in most medical imaging datasets, with entire MRI scans, eliminating the need for providing segmentation masks. In addition to improving the performance of pLGG molecular diagnosis, we examine the effect of this framework on the alignment between predictive imaging features and domain knowledge, as a measure of model explainability. Our results indicate a considerable improvement in the Area Under the Receiver Operating Curve compared to an MRI-only CNN model (0.863 vs 0.79), highlighting the value of radiology reports in enhancing CNN performance. However, the lower dice score between model attention maps and segmentation masks suggests that supervised training may not optimally integrate the two modalities. These findings signify the potential of multimodal learning in pLGG molecular diagnosis while underscoring the need for future exploration of other approaches, e.g., self-supervised learning, to improve the alignment between learnt features and clinical reasoning.

*Index Terms*—multimodal, brain MRI, radiology report, glioma, deep learning

## I. INTRODUCTION

Pediatric low-grade Glioma (pLGG) is one of the most common brain tumors in children, with a prevalence rate of 30-40%. Identifying the pLGG molecular subtypes or genetic markers can significantly aid in determining tumor severity and targeted treatment planning [3], [14]. Although recent research in radiology has led to the identification of some important imaging features corresponding to these genetic markers [14], diagnosing them from brain Magnetic Resonance Imaging (MRI) is typically beyond the purview of radiologists.

The current gold standard for identifying these genetic markers is through biopsy, which can come with several pitfalls. These include, but are not limited to, being invasive and costly as well as difficulty in accessing the tumor [16]. To address such challenges, MRI-based Convolutional Neural Networks (CNNs), a subset of Deep Learning (DL) and non-invasive diagnostic approaches, have been proposed and have demonstrated promising performance [8], [12]. Nonetheless, most of these models rely on regions of interest (ROIs) derived from manual segmentation masks, which are both labor-intensive and costly to obtain.

Radiology reports, on the other hand, are readily available and efficient data sources representing expert radiological knowledge. When integrated with MRI scans in a multimodal framework, these reports can improve the diagnostic performance of CNN models. They provide invaluable semantic context relevant to clinical tasks, which can guide the model. Moreover, these reports have the potential to assist the model in identifying and prioritizing informative features for a diagnostic task, i.e., enhancing model explainability.

In this work, we propose a multimodal MRI-report framework to improve the performance of pLGG genetic marker

classification. This framework combines context from both MRI and radiology report modalities in a supervised manner, leveraging complementary visual and textual information for the diagnostic task. Furthermore, we explore the essential textual features used by this framework, offering insights into how individual report words contribute to model decision-making.

To further assess the impact of radiology reports on model explainability, we compare the alignment of model attention maps with manual segmentation masks, as our major explainability metric. Finally, we evaluate the role of ROIs by training two additional frameworks: one based solely on ROIs and another on a combined ROI-report input, enabling a direct comparison between manually segmented inputs and report-driven guidance.

Our contributions in this paper can be summarized as follows:

- Developing three DL architectures for assessing the performance of MRI, radiology reports, and MRI-report concatenation in pLGG molecular diagnosis
- Extracting essential textual features contributing the most to the target predictions of the MRI-report model
- Evaluating the impact of MRI-report combination on the MRI-based model explainability
- Training two additional experiments on ROIs and ROI-report concatenation for comparing the impact of a CNN framework involving segmented inputs with that of multimodal report-based models

## II. LITERATURE REVIEW

Detecting the genetic markers of pLGG using ML has been explored in several works in the literature. Radiomics, i.e., extracting quantitative features from medical images [18], is one of the most frequently used methods. Wagner et al. [18] trained a random forest model on radiomics features for identifying the two most important pLGG genetic markers, i.e., BRAF Fusion and BRAF V600E Mutation, leading to an AUC of 85%. Vafaeikia et al. [15] developed a deep learning model for pLGG segmentation from MRI and applied a radiomics-based model to segmented tumor-related areas to classify its genetic markers, achieving an AUC of 84.3%. In spite of achieving promising results in these works, using radiomics features as input to ML algorithms typically requires explicit feature selection, a time-consuming process that may introduce bias into model predictions.

To address this issue and enable implicit feature selection, Convolutional Neural Networks (CNNs) have been explored for encoding MRI scans, which can automatically learn and extract relevant features. Namdar et al. [12] trained a CNN on a set of MRI scans and segmentation masks for binary pLGG genetic marker classification and achieved an Area Under the Receiver Operating Curve (AUC) of 86.11%. Similarly, Kudus et al. [8] developed a CNN-radiomics model for multi-class classification of the same task and obtained an AUC of 0.824.

Although these works have demonstrated promising performance in pLGG molecular diagnosis from MRI data, they typically rely on tumor segmentation masks provided by radiologists for extracting regions of interest (ROIs) in the image. Obtaining these masks, however, is a time-consuming and labor-intensive task, limiting the efficiency of such methods. In contrast, radiology reports serve as readily available data sources in most medical imaging datasets, imposing no additional costs on the healthcare system. These reports contain invaluable semantic information specified by radiologists related to medical imaging examinations, having the potential to improve diagnostic performance.

To the best of our knowledge, this is the first work that integrates radiology reports with entire MRI scans for pLGG molecular diagnosis. We propose a multimodal MRI-report framework to enhance the diagnostic performance. Furthermore, we employ gradient-based analysis to find the most informative textual features. Finally, we assess the impact of this framework on the visual explainability by comparing the attention maps of the image-based model with manual tumor segmentation masks.

## III. MATERIALS AND METHODS

### A. Data

Our dataset for this Research Ethics Board (REB)-approved study comes from a pediatric hospital in Canada. To the best of our knowledge, it is one of the largest and most unique datasets containing co-registered brain MRI scans (Fluid Attenuated Inversion Recovery (FLAIR) sequence), radiology reports, and genetic marker labels. For the course of this study, we used the cases associated with the two most prevalent genetic marker classes, namely BRAF Fusion and BRAF V600E Mutation, resulting in a total of 204 datapoints.

The images were preprocessed and resized to $240 \times 240 \times 155$ voxels, following the approaches illustrated in [9]. For preprocessing the reports, we removed all punctuation marks, numbers, dates, and phrases containing genetic marker-related information. Although the ground-truth genetic marker labels used in our study are derived exclusively from biopsy results, we found that some radiology reports included radiologists' preliminary impressions or suspected diagnoses related to specific genetic markers before observing biopsy results. These mentions reflect clinical speculation rather than confirmed genetic findings, and retaining them during training could lead to unintended data leakage. To prevent this, we carefully removed all explicit mentions of genetic markers from the text during preprocessing. These included exact terms and abbreviations such as BRAF fusion, BRAF V600E mutation, and similar phrases. These terms were identified based on a list of genetic markers relevant to our classification task.

To assess the effectiveness of the proposed framework in enhancing pLGG molecular diagnosis, we trained two baseline models on individual modalities, namely MRI scans and radiology reports, illustrated in Subsections III-B and III-C, respectively. The main multimodal framework is further explored in Subsection III-D. We used Adam optimizer [7], binary Cross-Entropy Loss, a batch size of 16, and 20 epochs.

We also employed StepLR scheduler to adjust the learning rate during training, reducing it by a factor of 0.5 every 10 epochs with a warm-up period of 10 epochs. We ran the experiments on a single "Tesla V100-PCIE-32G" GPU" using Pytorch 2. The code used for conducting the experiments is publicly available at https://github.com/IMICSLab/Supervised-Multimodal-Learning. Additionally, the dataset will be made available to researchers upon submission and approval of a data access request.

### B. MRI-based Classification

First, to investigate the predictive performance of MRI scans, we trained a 3D ResNet model using an initial learning rate of 0.0001. This model consists of a high number of parameters, and training it from scratch can lead to overfitting. To overcome this, we applied transfer learning by initializing the model's weights with those of a ResNet model pretrained on MedicalNet [2], called Med3D. MedicalNet is a large dataset containing diverse 3D medical images such as MRI, making it suitable for transfer learning for medical image analysis. After loading the pretrained weights, we froze the weights of the first two layers and fine-tuned only the last two ones. The reason is that in CNNs, first layers typically capture low-level features such as texture and edges, which are common among most diagnostic tasks, and last layers extract high-level and task-specific features.

In order to make the ResNet model assign higher weights to significant image areas, we utilized the attention mechanism [17]. It enables the model to emphasize relevant features while reducing the impact of less important areas. To that end, we incorporated a self-attention layer subsequent to the final convolutional module within the ResNet architecture. This layer contains three main components, namely query, key, and value. Based on the definition of self-attention, all these variables correspond to the same element of our model, which is the global representation of the model, i.e., the output of the last convolutional module.

The self-attention module operates by first transforming the three aforementioned components using a set of trainable parameters. The dot product of the transformed query and key forms the attention weights associated with the image representation. Subsequently, the weighted representation, added to the original one in a residual manner, gets forwarded to the FC layers for conducting the classification. The equations for finding the weighted representation can be found as follows:

$$\text{Query} = FW_Q$$
$$\text{Key} = FW_K$$
$$\text{Value} = FW_V$$
$$\text{Attention} = \text{softmax}\left(\text{Query}^T \times \text{Key}\right)$$
$$\text{Weighted Representation} = \text{Value} \times \text{Attention}^T$$

Where $F$ denotes the feature map extracted from the last convolutional layer; and $W_Q$, $W_K$, and $W_V$ are learnable parameters.

### C. Radiology Report-based Classification

In this experiment, we explored whether the interpretations provided by radiologists in radiology reports can be helpful in classifying pLGG genetic markers, although identifying these genetic markers is not within the purview of radiologists. To that end, we trained a language transformer, namely Longformer [11], on the MRI reports available in our dataset. Longformer [1] accepts sequence length of up to 4096 tokens, making it highly suitable for encoding long radiology reports. We initialized the weights of this transformer with Clinical Longformer [10], a variant of Longformer fine-tuned on MIMIC-III [4] clinical notes, which contains health-related information about patients admitted to critical care units. Similar to ResNet, we fine-tuned only the last two layers of this model while keeping the first ten layers frozen.

### D. Multimodal Framework

In this framework, we concatenated the MRI and radiology report representations to examine how well the combination of the two modalities can perform the classification task. We encoded the images and reports using 3D ResNet (with self-attention) and Clinical Longformer, respectively, to get the representation of each modality and then fused these representations.

Let:
- $\mathbf{x}^{(i)} \in \mathbb{R}^{d_i}$: Input MRI scan
- $\mathbf{x}^{(t)} \in \mathbb{R}^{d_t}$: Input radiology report
- $f_i(\cdot) : \mathbb{R}^{d_i} \to \mathbb{R}^{h_i}$: MRI encoder
- $f_t(\cdot) : \mathbb{R}^{d_t} \to \mathbb{R}^{h_t}$: Text encoder

The latent representations can be calculated as:

$$\mathbf{z}_i = f_i(\mathbf{x}^{(i)}) \in \mathbb{R}^{h_i}, \quad \mathbf{z}_t = f_t(\mathbf{x}^{(t)}) \in \mathbb{R}^{h_t}$$

These representations are concatenated and passed through a linear fusion layer for mapping them onto a joint embedding space:

$$\mathbf{z}_{\text{concat}} = [\mathbf{z}_i; \mathbf{z}_t] \in \mathbb{R}^{h_i + h_t}$$

$$\mathbf{z}_{\text{fused}} = \mathbf{W}_f \mathbf{z}_{\text{concat}} + \mathbf{b}_f, \quad \mathbf{W}_f \in \mathbb{R}^{h \times (h_i + h_t)}, \mathbf{b}_f \in \mathbb{R}^h$$

The fused representation is passed to a binary classification layer:

$$\hat{y} = \sigma(\mathbf{w}_c^\top \mathbf{z}_{\text{fused}} + b_c), \quad \mathbf{w}_c \in \mathbb{R}^h, b_c \in \mathbb{R}$$

$$where: \sigma(x) = \frac{1}{1 + e^{-x}} \quad \text{(sigmoid function)}$$

The model is trained using the binary cross-entropy loss:

$$\mathcal{L} = -y \log(\hat{y}) - (1 - y) \log(1 - \hat{y}), \quad y \in \{0, 1\}$$

The overall architecture of this framework, with 49,676,290 trainable parameters, is demonstrated in Figure 1.

To assess the contribution of each word to the final prediction, we applied a gradient-based explainability method called Integrated Gradients (IG) [13] to the input tokens, i.e., the subword units generated by the transformer's tokenizer, due to its implementation invariance. IG works by computing the path-integrated gradients from a baseline input (e.g., all zeros)

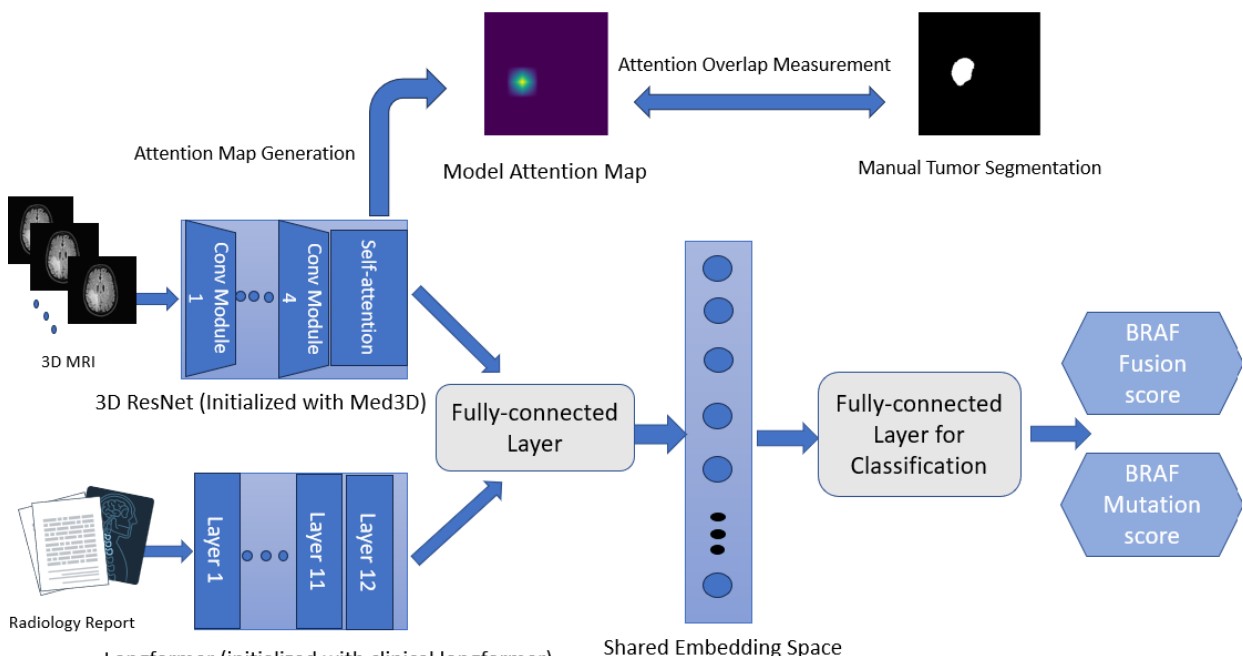

Fig. 1: Proposed Multimodal MRI-Report Framework: MRI and radiology reports are encoded by 3D ResNet and Clinical Longformer, respectively. Their representations are concatenated and mapped onto a shared embedding space via a fully connected layer, followed by another for classification. ResNet's attention maps are extracted and compared with segmentation masks for explainability assessment.

to the actual input, assigning an attribution score to each token. Higher scores indicate greater importance. To obtain word-level attribution scores, we averaged the attribution scores of the tokens corresponding to the same word.

### E. ROI-based Experiments

To evaluate the predictive value of radiology reports alongside imaging and compare it to ROI-based models where the ROIs were generated maually by a neuroradiologist, we conducted two additional experiments. The first focused solely on the tumor-related ROIs to determine whether integrating radiology reports could achieve comparable performance to ROl-based models. The second experiment was performed on the concatenation of ROIs and radiology reports to examine whether incorporating these reports can enhance ROI-based analysis, similar to the ROI-agnostic model.

### F. Visual Explainability Evaluation Setting

To determine the most important imaging areas contributing to a certain outcome, we visualized the attention weights obtained by the self-attention layer in the ResNet model in Experiments III-B and III-D. To preprocess these attention maps, we normalized their values to the range of [0,1] and binarized them using a threshold of 0.01, specified through visual assessment. Next, we computed the overlap between the preprocessed attention maps and manual tumor segmentation masks using Dice Coefficient, a metric used for image-to-image comparison. Higher dice scores indicate a better alignment between model attention and domain knowledge, indicating enhanced explainability.

### IV. RESULTS

We assess the classification performance and explainability of the developed experiments based on 5-fold cross-validation, enabling a robust evaluation of the classification outcomes. For each metric, we report the mean and standard deviation (Std) (in parentheses) on the test set, which remains independent of the training data across all folds.

### A. Classification Performance Evaluation

We used AUC as the main performance metric and applied precision, recall, and f1-score as minor metrics to have a more comprehensive assessment. Table I presents the mean values of these metrics obtained for the developed experiments.

A key observation is that our report-based model achieves significant performance, i.e., a mean test AUC of 0.845. This is higher than the AUC of the MRI-based model, which is 0.79, the lowest among all experiments. Furthermore, the proposed multimodal framework outperforms the two aforementioned models, obtaining an AUC of 0.863, which demonstrates the effectiveness of integrating image and text representations in the diagnostic task. This framework also achieves a statistically significant improvement in f1-score compared to the visual baseline (p-value=0.03). It leads to performance gains across most metrics, with the exception of recall, where the MRI-based model yields the highest value of 0.918 among the three experiments.

Evaluating the performance of the ROI-based and ROI-report architectures provides further insights into the impact of radiologist-provided data modalities on the classification performance. Notably, the ROI-based model does not significantly surpass the MRI-report framework, indicating the ability of our proposed framework to achieve results comparable to an ROI-based approach. Moreover, based on our primary metric, AUC, the combination of ROIs and reports achieves the highest score (0.9), signifying the critical role of radiology reports in enhancing the diagnostic performance when integrated with imaging data.

### B. Classification Explainability Evaluation

To assess the effect of the proposed multimodal framework on the visual explainability, we analyzed the attention maps generated by the MRI-based and MRI-report models. Table II represents the 2D and 3D dice scores, measuring the alignment between model-generated visual attention maps and segmentation masks. The 3D Dice score is computed across the entire MRI volume, while the 2D score is associated with the image slice having the largest cross-section with the tumor. These attention maps, along with the corresponding segmentation mask, have been depicted in Figure 3 for a shared datapoint in the test set. It is worth mentioning that since the segmentation masks serve as our explainability ground-truth, we did not use any ROI-driven models in this assessment.

In spite of enhancing the classification performance, the joint training of MRI scans and radiology reports within the multimodal framework reduces both 2D and 3D dice scores compared to the MRI-based model, thereby undermining the visual explainability. Furthermore, according to the provided visualizations, the MRI-report framework does not enhance the quality of model attention maps compared to the MRI-based model. This suggests that the supervised training of the two modalities may not be optimal for learning their representations as it misleads the visual attention. Instead, self-supervised approaches [6] can align the representations of the two modalities more effectively and lead to enhanced visual explainability.

Additionally, to explore the contribution of the textual features to pLGG molecular diagnosis, we extracted class-wise Word Cloud diagrams from the IG attribution scores, as described in Subsection III-D, which are depicted in Figure 2. As can be noticed, there are some relevant keywords, e.g., "cerebral" and "spinal" for BRAF Fusion and "inferior" and "frontal" for BRAF Mutation, among the informative report features. These can correspond to tumor location, an established clinical factor in pLGG genetic marker status [14].

### V. DISCUSSION

In this paper, we proposed a deep supervised multimodal framework that integrates brain MRI scans and radiology reports for pLGG genetic marker classification, an essential step toward improving tumor prognosis and informing treatment planning. In addition to evaluating the diagnostic performance of the framework, we assessed its visual explainability by

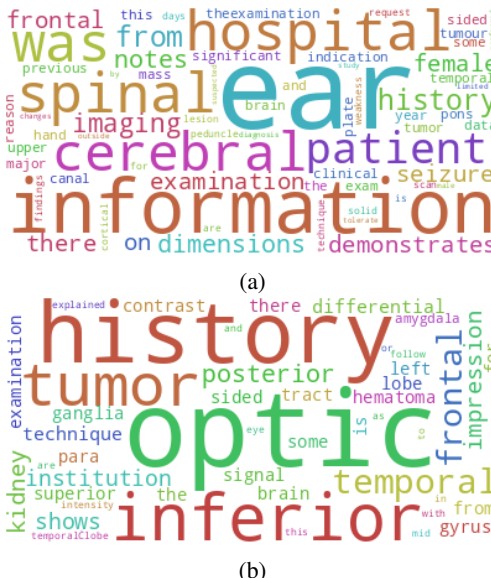

(a)

(b)

Fig. 2: Informative Textual Features for a: BRAF Fusion and b: BRAF V600E Mutation

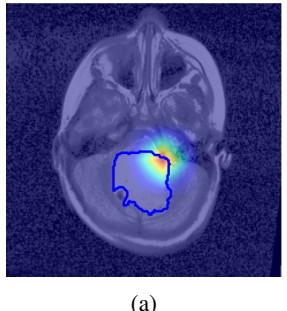 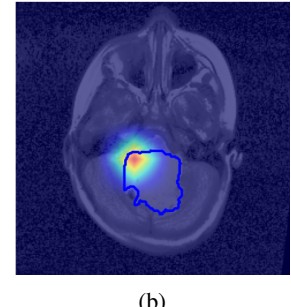

(a)                                    (b)

Fig. 3: MRI with overlaid attention map and manual tumor segmentation contours, for a test sample extracted from: a: MRI-based model and b: Multimodal model

measuring the overlap between the image-based model's attention maps and radiologist-provided segmentation masks. Furthermore, we applied gradient-based analysis to identify the most informative textual features used by the report-based model for detecting each genetic marker.

To have a comprehensive analysis of the predictive capacity of available data modalities, we conducted two additional experiments using segmentation-based ROIs and ROI-report combination. The results showed that fusing ROIs with radiology reports yielded the highest classification performance across all experiments, highlighting the strong diagnostic value of radiologist-derived data. Nevertheless, acquiring ROIs requires substantial human effort, limiting their scalibility in clinical workflows.

One of the most noteworthy findings emerged from the textual analysis. Although radiologists are not explicitly trained to diagnose pLGG genetic markers from MRI scans, the language they use in radiology reports can significantly assist

| Experiment | Mean AUC | Mean Precision | Mean Recall | Mean F1-score |
|---|---|---|---|---|
| Radiology Report | 0.845(±0.074) | 0.697(±0.215) | 0.631(±0.186) | 0.636(±0.150) |
| MRI | 0.79 (±0.101) | 0.380 (±0.064) | 0.918 (±0.086) | 0.534 (±0.068) |
| **MRI-Report** | 0.863(±0.115) | 0.858(±0.125) | 0.738(±0.225) | 0.758(±0.16) |
| ROI | 0.863(±0.097) | 0.464(±0.166) | 0.971(±0.035) | 0.609(±0.145) |
| **ROI-Report** | 0.9(±0.064) | 0.686(±0.17) | 0.687(±0.115) | 0.674(±0.12) |

TABLE I: Test Classification Performance of the Developed Models (Based on 5-fold Cross-validation)

| Experiment | 2D Dice Score | 3D Dice Score |
|---|---|---|
| MRI | 22.5% (±10.9%) | 8.9% (±4.8%) |
| MRI-Report | 15.9%(±12.2%) | 6.7%(±5.9%) |

TABLE II: ResNet Explainability Comparison between the MRI-based and MRI-report Multimodal Frameworks Based on Attention Overlap

in identifying these genetic markers. Moreover, by exploring the predictive textual features prioritized by the text-based model, we found that relevant keywords, e.g., those that potentially refer to tumor location, have gained significant attention by this model, highlighting the high impact of these features on model performance. Interestingly, these factors have been specified as clinically relevant to pLGG genetic markers in the radiological literature. This demonstrates the strong ability of our model to extract and leverage relevant semantic information from radiology reports and base its predictions on informative parts of these reports.

The integration of radiology reports with entire MRI scans, without relying on any ROIs, substantially enhanced the diagnostic performance, achieving a mean AUC of 0.863. This demonstrates the effectiveness of the proposed framework as an ROI-agnostic diagnostic tool. However, our findings also revealed a key limitation of supervised multimodal learning. Specifically, concatenating the two modalities under a classification loss undermined the visual explainability of the framework. This was evidenced by a reduction in the dice score between the imaging model's attention maps and ground-truth segmentation masks, indicating that the image model's focus was misdirected. A similar issue was reported in [5] for chest X-ray classification, where joint training of X-ray images with radiology reports under a single classification objective also led to degraded attention quality. This trade-off between classification performance and visual explainability highlights an important limitation of supervised multimodal learning in our setting. We believe this insight is valuable, as it motivates future work on alternative training strategies, beyond simple supervised fusion, for pLGG molecular diagnosis that better preserve modality-specific explainability in multimodal diagnostic systems.

An alternative approach is cross-attention-based fusion. Given the rich semantic content of radiology reports, the MRI representation could be treated as the query, and the report representation serve as the key and value. The output of this attention module would be then passed through a fully connected layer for classification. This architecture achieved a mean test AUC of 0.84, which is lower than that of the proposed framework (0.863). These results suggest that while cross-attention offers a more dynamic fusion strategy, its effectiveness may depend on dataset size and model complexity. The performance gap may be attributed to the additional trainable parameters introduced by the attention mechanism, which likely led to overfitting in our relatively small dataset.

As a future direction, we plan to replace the current supervised framework with a self-supervised architecture that learns the correlation between imaging and textual features during a pretraining phase using a much larger dataset, without relying on any classification loss at this stage. This framework operates by minimizing the distance between corresponding image-report pairs and maximizing the distance between mismatched pairs in a shared embedding space. Consequently, the model will encode image representations enriched with the semantic information contained in radiology reports. Following this pretraining phase, a downstream classifier can be trained solely on the learned image representations, without requiring any report integration. The embedded semantic knowledge is expected to improve both the classification performance and the visual explainability of the model. This separation between pretraining and classification offers a more explainable and efficient alternative, where radiologists can better understand the model's predictions, enabling its use as a reliable decision-support tool for pLGG genetic marker diagnosis.

## VI. CONCLUSION

In this work, we proposed a multimodal segmentation-free brain MRI-report framework for improving pLGG molecular diagnosis. Our evaluation, based on both classification performance and explainability, demonstrates a notable improvement in the performance when leveraging both modalities, yet at the cost of reduced visual explainability. This suggests that supervised multimodal learning may struggle to fully capture the underlying cross-modal correspondence, pointing to self-supervised learning as a more promising approach for aligning MRI and report representations.

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
