# OpenReview forum: "Exploring the Impact of Supervised Multimodal Learning on the Performance and Explainability of Pediatric Brain Tumor Molecular Diagnosis"
_IEEE.org/EMBS/BHI/2025/Conference — BHI 2025_

### Official Review · Reviewer_HGMi · 2025-07-15
**Exploring the Impact of Supervised Multimodal Learning on the Performance and Explainability of Pediatric Brain Tumor Molecular Diagnosis**

**Confidence:** 4
**Clarity Of Writing:** great
**Clinical Significance:** great
**Methodological Novelty:** great
**Overall Rating:** 7

**Experiments And Results:**

great

**Questions For The Authors:**

1. If radiology reports already summarize key findings, what is the added clinical value of incorporating MRI data into the multimodal model? Did you evaluate whether a text-only model (using reports) performs comparably to the multimodal model?
2. Have you tested the model on external datasets or planned steps to assess its robustness across different scanners and institutions?

**Strengths:**

1. Proposes a new ROI-free multimodal pipeline for pLGG using MRI and radiology reports in a supervised setting.
2. Demonstrated performance improvement using multimodal fusion (AUC 0.863) over MRI-only (AUC 0.79).
3. Explainability Analysis showed attention map alignment with tumor masks.

**Summary Of The Paper:**

This paper investigates a supervised multimodal deep learning method combining MRI scans and radiology reports for predicting genetic markers in pediatric low-grade glioma (pLGG).

**Weaknesses:**

Uses simple concatenation under a supervised classification loss which could be suboptimal for aligning image and text features (as authors acknowledge). Please Include these in the discussion section: how more advanced fusion strategies, such as cross-attention mechanisms, co-attention models, or transformer-based multimodal encoders, can be incorporated.

---

### Official Review · Reviewer_qgUr · 2025-07-16
**A multimodal clinical study with performance gains but limited novelty and questionable result reliability.**

**Confidence:** 5
**Clarity Of Writing:** great
**Clinical Significance:** good
**Methodological Novelty:** fair
**Overall Rating:** 5

**Experiments And Results:**

good

**Questions For The Authors:**

No.

**Strengths:**

1. The paper addresses a well-known clinical problem.
2. The methodology integrates state-of-the-art visual and language models
3. The experimental design is clear, and the paper is well-written.

**Summary Of The Paper:**

This paper focuses on classifying genetic markers in pediatric low-grade gliomas (pLGG) using brain MRI scans and radiology reports. It employs a ResNet-based model for MRI data (with optional ROI masks) and a transformer-based language model for radiology reports. The output embeddings from both modalities are concatenated and used for classification. Experimental results show that both radiology reports and ROI masks enhance classification performance over the baseline MRI model. The paper also explores explainability through visual attention maps and textual features.

**Weaknesses:**

1. The novelty is moderate. Multimodal architectures, especially vision-language models like CLIP, are widely used. The paper applies existing ideas and models without introducing new techniques or adaptations for the target domain.
2. The reliability of results is questionable. In Table 1, many standard deviations are large relative to the means. Additionally, the paper reports poor alignment between attention maps and tumor segmentation masks. These issues may indicate overfitting, data limitations, or suboptimal training. Without further analysis or clarification, the conclusions remain unconvincing.

---

### Official Review · Reviewer_fQKm · 2025-07-17
**Multimodal deep learning framework to predict genetic markers - strong methodology**

**Confidence:** 4
**Clarity Of Writing:** great
**Clinical Significance:** good
**Methodological Novelty:** great
**Overall Rating:** 7

**Experiments And Results:**

good

**Questions For The Authors:**

Can You Provide More Details on the Radiology Report Preprocessing? You mentioned “For preprocessing the reports, we removed all punctuation marks, numbers, dates, and words containing genetic marker-related information, which may be specified by radiologists as a preliminary diagnosis prior to receiving the biopsy result, to avoid data leakage.” Can you be more specific about “words containing genetic marker-related information”?

**Strengths:**

1. The paper addressed an important clinical issue of pediatric Low-Grade Glioma (pLGG). Being the current gold standard is biopsy, which is an invasive method, MRI, and report–based non-invasive prediction has clear clinical value.
2. The multimodal integration of MRI and report, combining 3D ResNet-based MRI image encoding with Clinical Longformer-based radiology report encoding, is a novel way of handling the problem.
3. Using real-world data adds credibility to the work.
4. The authors clearly acknowledged that visual explainability is degraded by multimodal fusion.

**Summary Of The Paper:**

This paper proposes a multimodal deep learning framework to predict genetic markers in pediatric low-grade glioma (pLGG) using brain MRI scans and radiology reports. They integrate A 3D ResNet-based encoder for MRI images (without segmentation masks), a Clinical Longformer-based encoder for radiology text, and a fusion module that jointly learns from both modalities to classify genetic status. The approach is trained and validated on a real-world clinical dataset of 204 datapoints from a pediatric hospital, with paired imaging, reports, and genetic testing. The model outperforms unimodal baselines in AUC, precision, and recall.

**Weaknesses:**

1. While AUC improves, Dice overlap between attention maps and tumor masks drops significantly in the multimodal settingwhich may raise concerns for clinicians who require spatial justification of model predictions.
2. No mention of public code or dataset, which might question the reproducibility.